# Changes in the Water Area of an Inland River Terminal Lake (Taitma Lake) Driven by Climate Change and Human Activities, 2017–2022

Feng Zi [1], Yong Wang [1,2,3,4], Shanlong Lu [2,3,5,*], Harrison Odion Ikhumhen [2,3], Chun Fang [2,3,5], Xinru Li [2,3,5], Nan Wang [2,3] and Xinya Kuang [2,3]

[1] School of Earth Sciences and Spatial Information Engineering, Hunan University of Science and Technology, Xiangtan 411201, China; zifeng@hnust.edu.cn (F.Z.); wyy2023@email.swu.edu.cn (Y.W.)

[2] International Research Center of Big Data for Sustainable Development Goals, Beijing 100094, China; harryspk@yhoo.com (H.O.I.); fangchun21@mails.ucas.ac.cn (C.F.); lixr2318@mails.jlu.edu.cn (X.L.); 107552104290@stu.xju.edu.cn (N.W.); s6xikuan@uni-bonn.de (X.K.)

[3] Key Laboratory of Digital Earth Science, Aerospace Information Research Institute, Chinese Academy of Sciences, Beijing 100094, China

[4] School of Geographical Sciences, Southwest University, Chongqing 400715, China

[5] University of Chinese Academy of Sciences, Beijing 100049, China

* Correspondence: lusl@aircas.ac.cn

**Abstract:** Constructed from a dataset capturing the seasonal and annual water body distribution of the lower Qarqan River in the Taitma Lake area from 2017 to 2022, and combined with the meteorological and hydraulic engineering data, the spatial and temporal change patterns of the Taitma Lake watershed area were determined. Analyses were conducted using Planetscope (PS) satellite images and a deep learning model. The results revealed the following: ① Deep learning-based water body extraction provides significantly greater accuracy than the conventional water body index approach. With an impressive accuracy of up to 96.0%, UPerNet was found to provide the most effective extraction results among the three convolutional neural networks (U-Net, DeeplabV3+, and UPerNet) used for semantic segmentation; ② Between 2017 and 2022, Taitma Lake's water area experienced a rapid decrease, with the distribution of water predominantly shifting towards the east–west direction more than the north–south. The shifts between 2017 and 2020 and between 2020 and 2022 were clearly discernible, with the latter stage (2020–2022) being more significant than the former (2017–2020); ③ According to observations, Taitma Lake's changing water area has been primarily influenced by human activity over the last six years. Based on the research findings of this paper, it was observed that this study provides a valuable scientific basis for water resource allocation aiming to balance the development of water resources in the middle and upper reaches of the Tarim and Qarqan Rivers, as well as for the ecological protection of the downstream Taitma Lake.

**Keywords:** climate change; human activities; Planetscope; water area; deep learning; Taitma Lake

## 1. Introduction

Lakes are crucial freshwater bodies that serve significant roles in biogeochemistry and the hydrological cycle, and are vital natural resources necessary for human socio-economic development [1,2]. The surface area of a lake is primarily determined by two components: water input (e.g., precipitation, surface inflow, groundwater input) and water output (e.g., evapotranspiration, surface outflow, groundwater output), which are influenced by climate change and human endeavors [3]. In the context of global warming, the arid and semi-arid regions of northwestern China have undergone an abrupt climate shift from "warm-dry" to "warm-wet" [4], resulting in dramatic changes in the number and area of numerous inland lakes in China [5]. These alterations will negatively impact their sustainable economic and

social development, especially in the already environmentally vulnerable oasis ecosystems of the lower inland rivers of the arid zone.

Climate change and human activities continuously reshape the spatial positions and morphologies of lake water bodies, thereby necessitating real-time, dynamic monitoring [6]. Remote sensing technology facilitates this with its broad, dynamic, and cost-effective monitoring capabilities, enabling rapid and precise acquisition of lake surface data at high temporal frequencies, thus supporting the monitoring of spatiotemporal changes in lake extents [7]. Remote sensing methods for water body extraction are divided based on pixel spectral characteristics and spatial textures into single-band thresholding, water index methods, object-oriented classification, and machine and statistical learning approaches [8,9]. Among these, machine and statistical learning are distinguished by their ability to simulate and learn from data features, achieving high accuracy in water identification. Deep learning models like U-Net, DeepLabv3+, and UPerNet greatly enhance processing efficiency and accuracy. U-Net effectively delineates water boundaries against complex backgrounds through its symmetric structure and feature fusion [10]. DeepLabv3+ processes multi-scale information using atrous convolution, ideal for post-disaster water body detection in complex terrains [11]. UPerNet identifies water bodies in small and complex areas with pyramid pooling and feature fusion [12]. These models not only refine the precision of remote sensing image processing but also improve monitoring of dynamic water body changes.

Taitma Lake, also known as Kharabrang Haiji, is the largest inland river in China and the tail lake of the Tarim and Qarqan rivers. It serves as a crucial component of the aquatic ecosystem and an indicator of river health, while also holding special ecological significance and resource potential [13,14]. Until 1964, the Tarim River consistently supplied Taitma Lake with a substantial volume of water, and the Qarqan River annually received a modest contribution from the Tarim. Since the 1970s, however, the interception at the Daxihaizi Barrage Reservoir and increased socio-economic water use in the upper reaches have led to a gradual desiccation of Taitma Lake and severe degradation of the area's natural environment, causing widespread concern among the public and government [15,16]. To reverse this trend, since May 2000, the local government has implemented the Tarim River ecological water transfer project. With the progression of this ecological intervention, the region's lake areas have seen significant alterations, the groundwater has risen, the water quality has improved, and the ecological environment has been significantly enhanced [17–19]. This dynamic of "ecological water transfer—lake area evolution—ecological environment response" has garnered scholarly attention. While previous studies have largely concentrated on the ecological responses and groundwater levels in the lower reaches of the Tarim River [20], substantial changes in the water area of Taitma Lake at the downstream end, particularly following recent extensive water conservation projects and upstream management alterations, have been largely neglected. Traditional satellite imagery such as Landsat and Sentinel, with spatial resolutions of 30 m and 10 m, respectively, fails to effectively monitor the narrow, seasonal water bodies at the Taitma Lake inlet. To address these challenges, this paper utilizes high spatial resolution (3 m), high temporal frequency (daily) PlanetScope (PS) satellite imagery combined with a deep learning semantic segmentation model to accurately delineate fine water bodies in this region over the past six years. Moreover, this study investigates the impacts of climate change and human activities on the Taitma Lake watershed in conjunction with meteorological data. The primary objectives of this research are to evaluate the effectiveness of using high-resolution commercial satellite data (PS) and deep learning models for monitoring changes in the distribution of small- and medium-sized river water bodies in arid and semi-arid regions, and to elucidate the patterns and drivers of spatiotemporal change in the Taitma Lake water area from 2017 to 2022.

## 2. Materials and Methods

### 2.1. Study Area

Taitma Lake is located in the Luobuzhuang region of Xinjiang Uygur Autonomous Region, China, approximately 100 km north of Ruoqiang County, adjacent to the Takla-makan and Kumtag deserts (Figure 1). This area is characterized by extreme temperatures and scant precipitation throughout the year, reflecting a harsh dry continental climate. The average annual precipitation ranges from 17.4 to 42.0 mm, while the average annual temperature is greater than 10 °C. The annual potential evaporation is notably high, ranging from 2500 to 3000 mm, marking it as one of the driest regions in China. The vegetation's water supply is almost entirely reliant on groundwater replenishment [21]. Its surface water sources mainly come from the Tarim River and Qarqan River in the "nine sources and one dry" river in the Tarim River Basin [22]. In 1972, a combination of climate change, river diversion, and excessive water use led to the degradation of the lower Tarim River and the desiccation of Taitma Lake [23]. To safeguard the local ecosystem, the Chinese government allocated CNY 10.7 billion in 2000 to initiate the ecological water delivery project. Up to now, the water area of Taitma Lake has gradually recovered, the groundwater level has risen significantly, and the ecological management practices along the riverbanks have yielded impressive outcomes [24,25].

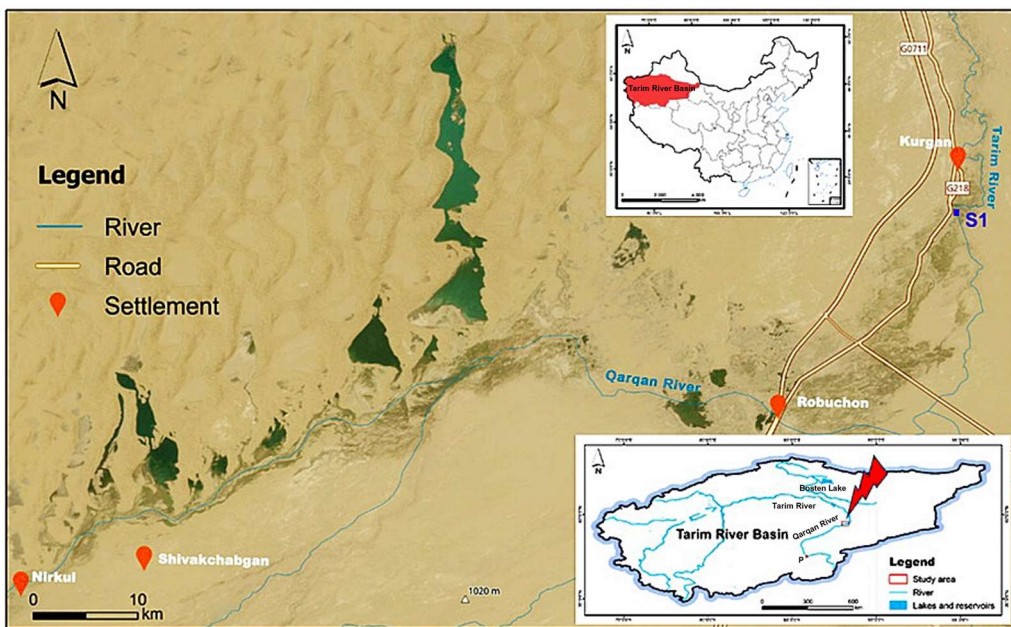

**Figure 1.** Location of the study area.

### 2.2. Data Sources

The data analyzed in this paper are composed of two main parts: (1) surface water remote sensing monitoring data; (2) meteorological and other auxiliary data.

(1) Surface water remote sensing monitoring data. These mainly consist of PS satellite imagery and a global monthly water surface distribution dataset (JRC Monthly Water History, v1.4). The former is derived from Planet's official website (https://www.planet.com/, accessed on 13 March 2023). PS satellite products only have 4 multispectral bands of visible light (RGB) and near-infrared (NIR), with a band range of 455–860 m, a temporal resolution of 1 day, and a spatial resolution of up to 3 m [26]. However, PS satellite imagery only became accessible on 22 June 2016, and was subject to considerable data acquisition constraints. Specifically, the Planet website offers each user a monthly download limit of 5000 square kilometers, with each image covering a swath of only 24.6 km by 16.4 km. As a result, we used PS imagery exclusively to track the seasonal changes in water coverage of Taitma Lake from 2017 to 2022. The latter comes from the GEE (Google

Earth Engine) platform with a temporal resolution of 1 month and a spatial resolution of 30 m (https://global-surface-water.appspot.com/, accessed on 24 March 2023). The dataset was extracted using 4,453,989 images of Landsat 5, 7, and 8 acquired during the period 16 March 1984, to 31 December 2020 [27]. To maximize the utilization of PS imagery and minimize data wastage, this research extracted monthly surface water distribution data from the Taitma Lake area for the period from 2017 to 2022. These data were used to identify the seasonal dates of the maximum catchment area of Taitma Lake, which were then aligned with the seasonal collection dates of the PS imagery.

(2) Meteorological and other supplementary data. Meteorological data were obtained from the China Meteorological Science Data Sharing Network (http://data.cma.cn/, accessed on 27 May 2023), including monthly precipitation, temperature, and evapotranspiration data for Ruoqiang and Qiemo counties from 2017 to 2022. Other auxiliary data used in this study include datasets on the Tarim River Basin boundaries, distribution of rivers, lakes, and reservoirs from the National Tibetan Plateau Science Data Center (https://data.tpdc.ac.cn/, accessed on 16 February 2023), and Tarim River ecological water transport data from the Tarim River Basin Management Bureau (http://www.tahe.gov.cn/, accessed on 12 June 2023).

### 2.3. Methodology

In this study, PS satellite images served as the primary data source, and platforms and tools such as GEE, ArcGIS, and Python were extensively employed to successfully construct the fine water body distribution dataset in the study area as well as the river widths of the river cross sections at the inlet of Taitma Lake from 2017 to 2022. Based on this, the spatial and temporal characteristics of the watershed area of Taitma Lake affected by the effects of climate change and anthropogenic activities were examined from 2017 to 2022 using the methods of M–K trend, Pearson's correlation, and the center of gravity analytical model. The meteorological data from stations in Ruoqiang and Qiemo counties as well as data from hydraulic hub projects in the area were integrated with these models (Figure 2).

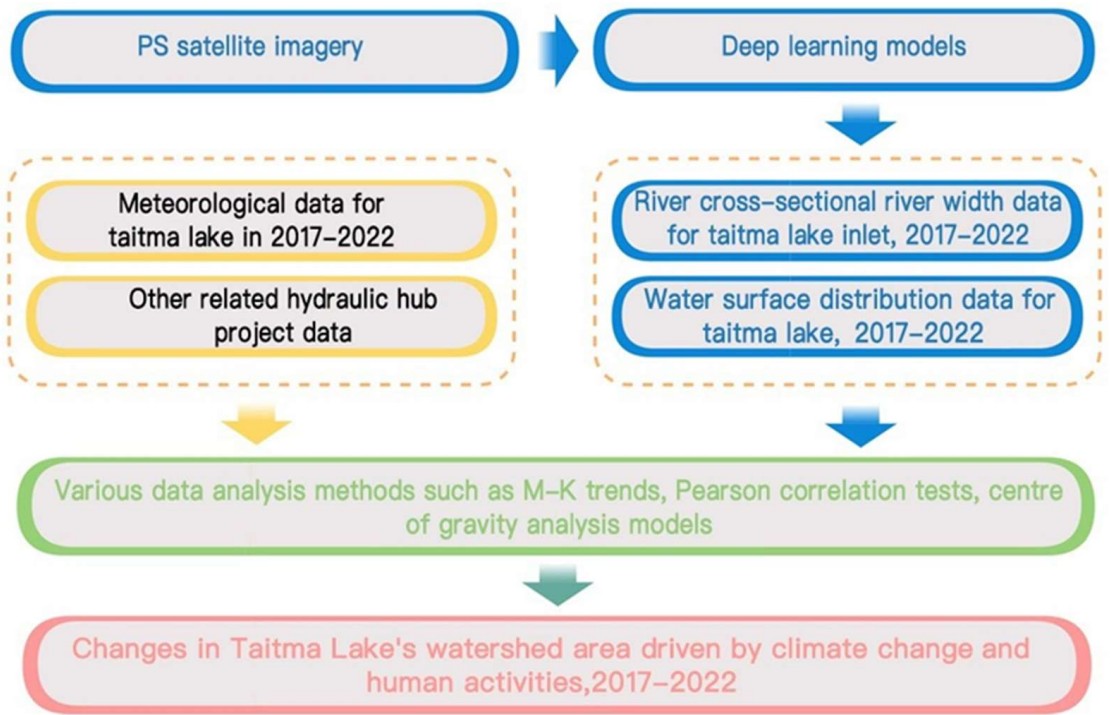

**Figure 2.** Research method flow.

2.3.1. Water Extraction and Evaluation Methods

(1)    Image data acquisition

In this study, seasonal water body distribution data for Taitma Lake from 2017–2022 were extracted from PS satellite images based on a deep learning model to analyze the spatial and temporal patterns of water surface area in the region. To improve the robustness and generalization capabilities of the algorithmic model, we collected 1056 PS images of 512 × 512 pixels as the experimental dataset in 25 uniformly selected sampling regions in the Tarim River Basin (Figure 3). The principles of sample collection include the following: ① For different sensor types, the data cover three years of 2017 (PS2), 2019 (PS2.SD), and 2021 (PSB.SD) (the radiation resolution of PS images from different sensors is 16 bits); ② Due to the complex environment around water bodies, the spectral and texture characteristics of water bodies in different regions are quite different. In order to test the universality of these network models for water body extraction, we considered environmental features such as spectrum, texture, and seasons in the dataset (mainly March and September) and water environment characteristics (different types of water bodies, such as urban water bodies, plains-type river reservoirs or lakes, and plateau-type river reservoirs or lakes, etc.).

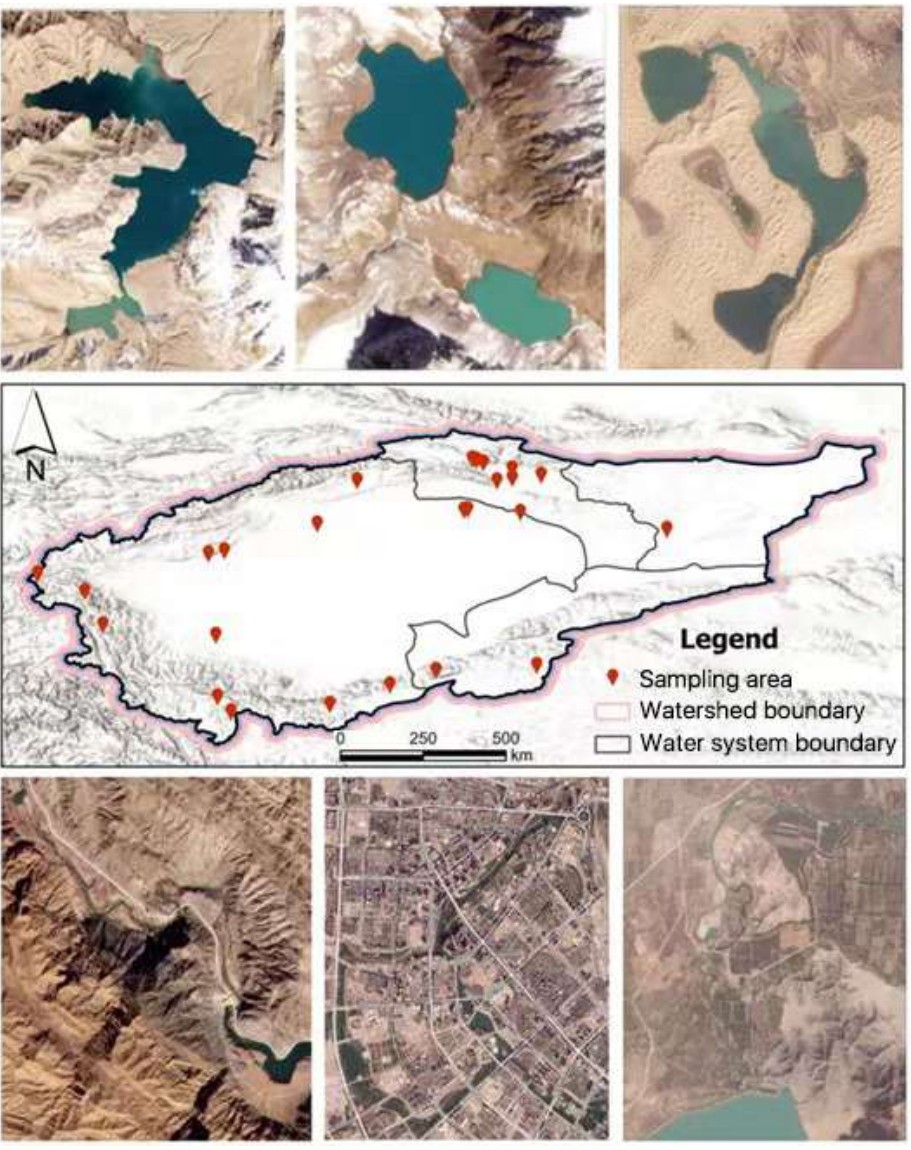

**Figure 3.** PS image dataset and geographical distribution.

(2)    Water extraction process

The acquisition process of the water body distribution dataset in the Taitma Lake area is mainly divided into four parts: image preprocessing, sample generation, water body extraction, and accuracy evaluation (Figure 4). To reconcile computational demands with the capability of effectively capturing image features in deep semantic segmentation models such as U-Net, UPerNet, and DeepLabv3+, we segmented selected PlanetScope satellite imagery into 512 × 512 pixel tiles using a Python script [28,29]. This ensured that the models efficiently process high-resolution inputs while maintaining manageable computational loads, thereby optimizing performance and resource utilization. In the second part, the cropped remote sensing image samples were manually labeled with water bodies, and the water body samples were randomly divided into training set, validation set, and test set according to the ratio of 8:1:1 and the type of region, and at the same time ensured that the three types of samples could exist simultaneously and covered almost the same area in each sampling region. The third step used methods such as U-Net, DeepLabv3+, and UPerNet for model training to extract water bodies from PS satellite images. Finally, the accuracy of different methods was evaluated using visual comparison and quantitative evaluation indicators to determine the optimal method for PS satellite image water extraction.

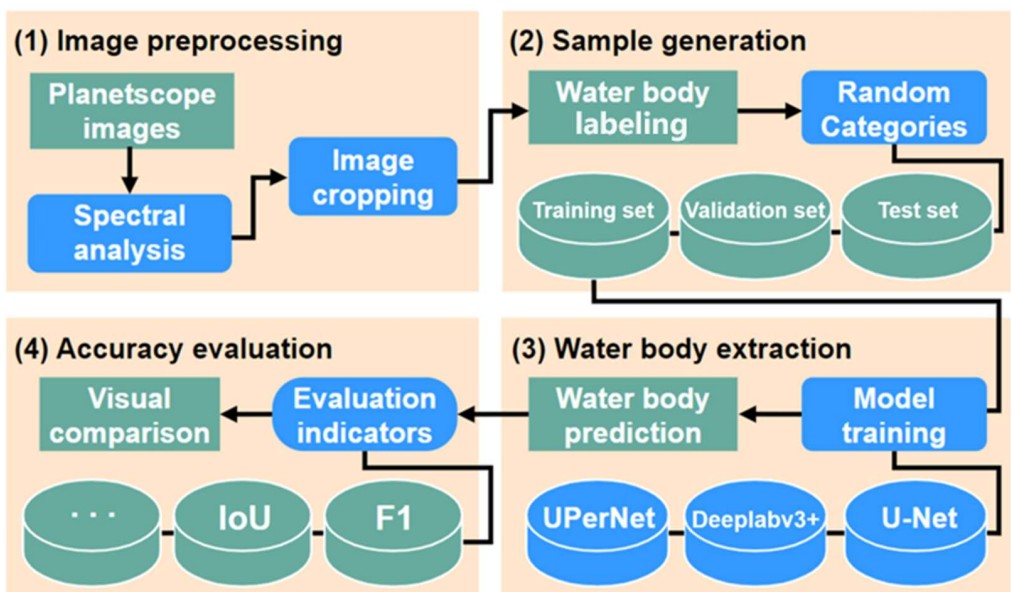

**Figure 4.** Fine water body distribution data acquisition process.

(3)    Accuracy evaluation index

The precision (P), recall (R), F1 value (F1), pixel accuracy (PA), and intersection over union (IoU) as accuracy evaluation indicators were primarily obtained for the fine water body distribution data extracted in this paper using the same evaluation method as the classification of remote sensing images. The specific formula is as follows:

$$P = \frac{p_{00}}{p_{00} + p_{10}} \tag{1}$$

$$R = \frac{p_{00}}{p_{00} + p_{01}} \tag{2}$$

$$F1 = \frac{2 \times P \times R}{P + R} \tag{3}$$

$$PA = \frac{p_{00} + p_{11}}{p_{00} + p_{01} + p_{10} + p_{11}} \tag{4}$$

$$\text{IoU} = \frac{p_{00}}{p_{00} + p_{01} + p_{10}} \tag{5}$$

where $p_{00}$ denotes the number of pixels originally classified as water bodies and correctly predicted as water bodies, $p_{01}$ denotes the number of pixels originally classified as water bodies but incorrectly predicted as non-water bodies, $p_{10}$ denotes the number of pixels originally classified as non-water bodies but incorrectly predicted as water bodies, and $p_{11}$ denotes the number of pixels originally classified as non-water bodies and correctly predicted as non-water bodies.

2.3.2. Data Analysis Methods

(1)  Mann–Kendall trend test

The Mann–Kendall ("M–K") test is a nonparametric test [30,31] proposed by H.B. Mann and M.G. Kendall, which is recognized as a standard nonparametric statistical test for trend detection of variables at different scales. In the M–K test, the original hypothesis $H_0$ states that there is no trend in the sample series. Given a time series data being $(X_1, X_2, X_3, \ldots, X_n)$, then the tau formula that measures the magnitude of the trend in the rejection of the original hypothesis $H_0$ is as follows:

$$\text{tau} = \text{Median}\left(\frac{x_k - x_i}{k - i}\right), \ \forall \, i < k \tag{6}$$

In the formula, for $1 < i < k < n$, $x_i$ and $x_k$ are respectively defined as the i-th and k-th data values in the time series. When tau > 0, it indicates that the sample has an increasing trend; when tau is negative, it indicates that there is a decreasing trend.

(2)  Pearson correlation test

The covariance (COV) is a prerequisite for calculating Pearson's correlation coefficient (COR), which numerically characterizes the interrelationship between two variables $(x,y)$. This calculation also requires that the number of observations n be greater than 2; the COV calculation formula is as follows:

$$\text{COV}(X, Y) = \frac{1}{n - 1}\sum\nolimits_{i=1}^{n}\left(X_i - \overline{X}\right)\left(Y_i - \overline{Y}\right) \tag{7}$$

In the formula, $X_i$ and $Y_i$ are the actual data values in the two groups of variables x and y, and $\overline{X}$ and $\overline{Y}$ are the average values of $X_i$ and $Y_i$, respectively. Generally speaking, the range of COV values is inconsistent and affected by the dimension, while COR can use the covariance divided by the standard deviation to eliminate the influence of dimension [32]. The calculation formula of COR$(x,y)$ is as follows:

$$\text{COR}(x, y) = \frac{\sum_{i=1}^{n}\sum_{j=1}^{n}(x_i - \overline{x})(y_i - \overline{y})}{\sqrt{\sum_{i=1}^{n}\sum_{j=1}^{n}(x_i - \overline{x})^2(y_i - \overline{y})^2}} \tag{8}$$

where $I, j = 1, 2, \ldots n$, the value range of COR$(x,y)$ is $[-1, 1]$, and the closer the value of $|\text{COR}(x,y)|$ is to 1, the stronger the correlation between the variables x and y. Conversely, the closer the value is to 0, the weaker the correlation, with 0 indicating no correlation. Meanwhile, in the output Pearson table, if the upper right mark of the COR value is a single asterisk, it signifies that the one-sided test is satisfied, and a double asterisk means that the two-sided test is satisfied. If the correlation coefficients of the corresponding indicators in the table of values do not have an asterisk, it is considered that it has not passed the significance level test, suggesting that the two related variables do not have correlation.

(3)  Center of gravity analysis model

The principles, concepts and calculation formulas involved in the water body area center of gravity analysis model are shown by Equations (9)–(12).

First, the formula for calculating the geometric center of gravity is as follows:

$$\overline{x} = \frac{\sum_{i=1}^{n} x_i}{n}, \ \overline{y} = \frac{\sum_{i=1}^{n} y_i}{n} \tag{9}$$

where $n$ is the total number of spatial objects, $x_i$, $y_i$ are the coordinate values of the ith spatial unit, and $\overline{x}$, $\overline{y}$ are the coordinate values of the geometric center. The water body area center of gravity model equation is as follows:

$$X = \frac{\sum_{i=1}^{n} s_i \cdot \overline{x_i}}{\sum_{i=1}^{n} s_i}, \ Y = \frac{\sum_{i=1}^{n} s_i \cdot \overline{y_i}}{\sum_{i=1}^{n} s_i} \tag{10}$$

where $X$, $Y$ are the longitude and latitude values of the center of gravity coordinates of the area attribute, respectively, $s_i$ represents the area of the i-th water body area, $\overline{x_i} \ \overline{y_i}$ are the coordinates of the geometric center of the ith water body region, and n is the number of water body regions. The direction of inter-annual spatial movement of the center of gravity is calculated as follows:

$$\theta_{j-i} = \frac{\pi}{2} + arctan\frac{y_j - y_i}{x_j - x_i}(n = 0, 1, 2) \tag{11}$$

where $\theta_{j-i}$ denotes the angle at which the center of gravity moves from year i to year j ($-180° \leq \theta_{j-i} \leq 180°$), with counterclockwise rotation being positive and clockwise rotation being negative. When the center of gravity shifts to the northeast, $0° < \theta < 90°$; when the center of gravity shifts to the northwest, $90° < \theta < 180°$; when the center of gravity shifts to the southwest, $-180° < \theta < -90°$; when the center of gravity shifts to the southeast, $-90° < \theta < 0°$; when the center of gravity shifts to the due east or the due west, $\theta = 0°$ or $\theta = \pm180°$; and when the center of gravity shifts to the due north or the due south, $\theta = \pm90°$. The center of gravity inter-annual spatial movement distance is calculated as follows:

$$D_{j-i} = c\sqrt{\left(X_j - X_i\right) + \left(Y_j - Y_i\right)} \tag{12}$$

where $D_{j-i}$ denotes the distance that the center of gravity moves from year i to year $j$, and c is the coefficient of the transformation of the geographic coordinate system (°) into the plane projection coordinate system (km), c = 111.111.

(4) Annual maximum water surface synthesis

The annual water body distribution dataset for Taitma Lake is derived through a synthesis of the maximum seasonal water body extents within the year for the specified region. The computation is conducted using the following formula:

$$ASW = MAX\{msw_1, msw_2, msw_3, msw_4\} \tag{13}$$

where ASW is the annual water surface data (km$^2$), and $msw_1$, $msw_2$, $msw_3$, and $msw_4$ represent the maximum water surface distribution data (km$^2$) in spring, summer, autumn, and winter of the year, respectively.

## 3. Results

### 3.1. Evaluation of the Accuracy of Water Extraction Results

In order to select the optimal water body extraction method, a comparative experiment was carried out between the existing common deep learning segmentation models (U-Net, Deeplabv3+, and uPerNet) and the normalized difference water index threshold method (NDWI). Moreover, analysis based on the extraction accuracy of the overall water body and the fine water body was undertaken to clearly illustrate the variations in the extraction accuracy of different water body types by various techniques.

3.1.1. Accuracy Evaluation Results of the Overall Water Body

The extraction results of the entire water body demonstrate that the deep learning-based water body identification approach outperformed the conventional method, with the mIoU increasing from 7% to 12%. Among the three semantic segmentation models, uPerNet has the best water extraction effect, with an accuracy P of 96.0%, followed by Deeplabv3+, and U-Net exhibiting the least effective performance. Compared with U-Net and Deeplabv3+, UperNet's F1 value increased by 2.6% and 0.9%, respectively, and the mIoU increased by 4.8% and 1.6%, respectively, which shows that uPerNet can effectively solve the problems of low resolution and high noise in water body labels based on PS satellite images that affect the accuracy of water body extraction (Table 1). The visual comparison of the water body extraction results revealed that, except for UperNet, the findings generated by the other three techniques exhibited some degree of misclassification. This indicates that the uPerNet segmentation model could accurately classify both water bodies and non-water bodies (Figure 5). Reasons for this performance include the following: U-Net fuses features at multiple scales through skip connections between decoders and encoders, which facilitates more accurate extraction of water body boundaries and captures detailed information in images [33]. However, U-Net incorporates too many low-level features extracted by shallow convolutional layers, and these low-level feature maps may be mixed with a lot of noise that has similar spectral characteristics to water bodies, thus causing U-Net's water body extraction effect to be less effective. Conversely, Deeplabv3+ is one of the most advanced convolutional neural networks (CNN) in the field of computational vision, which is able to extract features at multiple scales using atrous spatial pooling pyramid (ASPP) and restore the resolution of feature maps using decoders [34]. The poor performance of Deeplabv3+ in this study may be related to its complex structure [35]. It may be suitable for pixel segmentation in complex scenes, but over-fitting is prone to occur in water extraction.

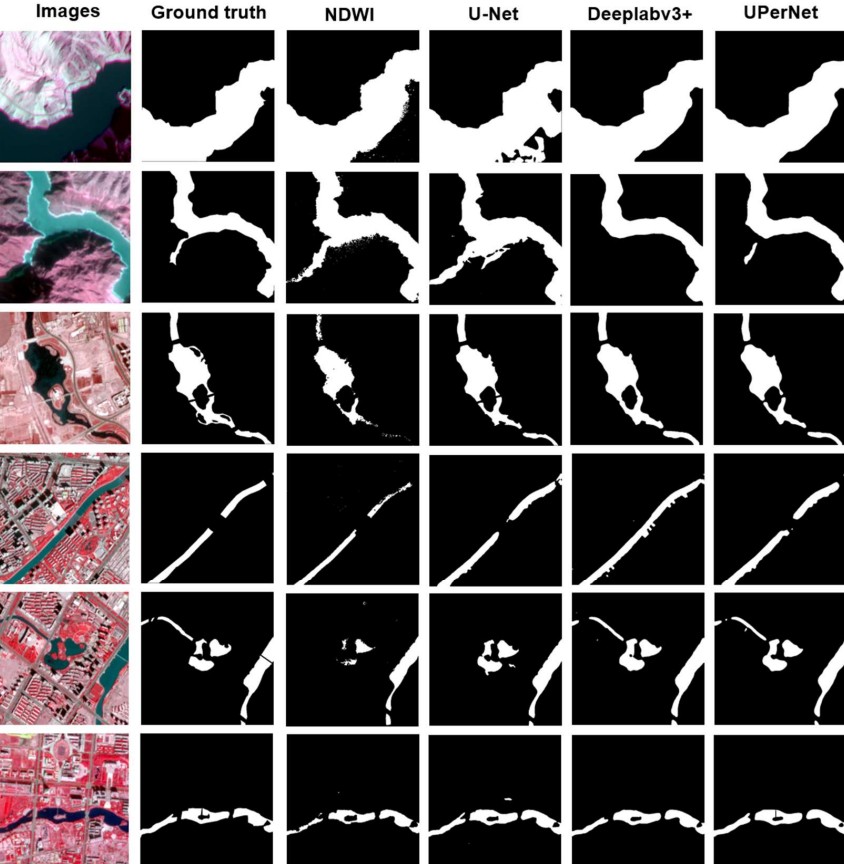

**Figure 5.** Comparison of water extraction results of different methods.

**Table 1.** Comparison of water body extraction accuracy with different methods.

| Method | P (%) | R (%) | F1 (%) | PA (%) | mIoU (%) |
|--------|-------|-------|--------|--------|----------|
| NDWI | 85.4 | 94.7 | 89.8 | 93.2 | 81.4 |
| U-Net | 90.4 | 97.8 | 94.0 | 96.1 | 88.6 |
| Deeplabv3+ | 94.2 | 97.3 | 95.7 | 97.3 | 91.8 |
| uPerNet | 96.0 | 97.1 | 96.6 | 97.9 | 93.4 |

3.1.2. Accuracy Evaluation Results of Fine Water Bodies

In order to compare the extraction effects of the four methods on water bodies like fine rivers and lakes, and to provide quantitative analysis results, we selected 144 samples from the original set that contained only fine water bodies. This selection was aimed at studying the dynamics of fine water bodies within this research. The experimental results demonstrated that although the four methods decreased the extraction accuracy of fine water bodies compared to overall accuracy, the deep learning-based water body detection method significantly outperformed the conventional water body index method in terms of extraction efficiency. Compared to uPerNet, misclassification was particularly noticeable with other methods, especially the water body index and U-Net, and significant differences in accuracy were observed. Although the uPerNet method performed poorly on small rivers and lakes, it was capable of producing excellent results (P at 90.0%, F1 as high as 90.2%, and mIoU at 82.2%) (Table 2, Figure 6). The possible reason is that the limited effectiveness of U-Net and Deeplabv3+ in distinguishing urban building shadows from water bodies significantly impacts their water extraction accuracy. U-Net struggles with accurate differentiation, while Deeplabv3+, despite its superior noise suppression capabilities, faces challenges in detailing small and fine features due to the 3 m resolution constraints of PS satellite imagery [11,36]. Most fine water samples in this study are from urban areas, where this resolution limitation is particularly problematic. Consequently, uPerNet surpasses both models, demonstrating enhanced accuracy with superior performance metrics, suggesting its architecture is better suited for detailed urban water body extraction.

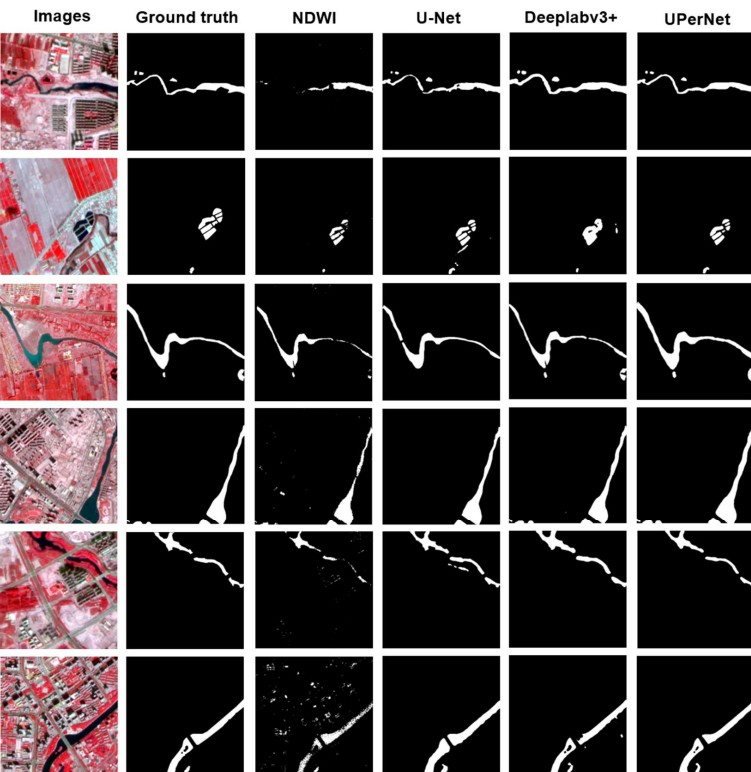

**Figure 6.** Comparison of extraction results of fine water bodies by different methods.

**Table 2.** Comparison of extraction accuracy of fine water bodies by different methods.

| Method | P (%) | R (%) | F1 (%) | PA (%) | mIoU (%) |
|---|---|---|---|---|---|
| NDWI | 86.5 | 54.9 | 70.0 | 89.6 | 53.8 |
| U-Net | 84.3 | 76.1 | 80.0 | 98.5 | 66.7 |
| Deeplabv3+ | 75.4 | 97.4 | 85.0 | 98.6 | 73.9 |
| UPerNet | 90.0 | 90.5 | 90.2 | 99.2 | 82.2 |

*3.2. Temporal and Spatial Changes in the Water Area of Taitma Lake*

This section primarily uses the optimal method (UPerNet), identified from previous experiments on PS satellite image water extraction, to analyze the seasonal variations in the surface area of Lake Taitma from 2017 to 2022. The results show that the UPerNet method consistently exceeds the results obtained from JRC products, largely due to the high resolution (3 m) of PS images and the enhanced water body detection capability provided by optimized water extraction methods (Figure 7). Our analysis clearly reveals that the water area of Taitma Lake displays pronounced seasonal fluctuations over temporal scales. Contrary to expectations, the maximum annual water surface area typically occurs in March, a spring month, rather than during the summer months. Moreover, over the past six years, the largest water area was recorded in 2019 at 391.33 km$^2$, followed by 2017 with 384.527 km$^2$. The smallest annual water areas were observed in 2021 and 2022, at 139.28 km$^2$ and 143.72 km$^2$, respectively. Upon transforming the seasonal water surface areas of Taitma Lake into annual distribution data using Equation (13), our analysis over the past six years indicates Taitma Lake's water area passed the M–K trend test, with a clear downward trend (Kendall's tau value is −0.733) and a significant decline at an annual average rate of 31.12 km$^2$, with a clear turning point shift in 2020 (Figure 7, Table 3). Spatially, the center of gravity of water distribution of Taitma Lake in 2017–2022 was located between 88.099°E~88.163°E, 39.509°N~39.511°N. Compared to the lake's geometric center of gravity in present-day Ruoqiang County, southeastern Bayingolin Mongol Autonomous Prefecture, Xinjiang (88.044°E, 39.561°N), there has been some discrepancy towards the east and south of the center of gravity of the distribution of its water bodies, which is related to the ecological water transfer project implemented by the local government since 2000, with the source of the recharge water of the lake (the Tarim River) mostly concentrated in the southeastern region of the Taitma Lake area (Figure 1). From the overall trend of the center of gravity, over the past 6 years, the center of gravity of the water distribution of Taitma Lake has moved about 7.24 km to the northwest, while the center of gravity has moved 7.20 km in the east–west direction, with an average annual movement of 1.20 km; the center of gravity in the north–south direction only moved 0.68 km, with an average annual movement of 0.11 km. It is also evident that during the last six years, the rate of movement of the center of gravity of Taitma Lake's water distribution in the east–west direction has been significantly greater than in the north–south direction. This is likely related to the hydration state of the east–west flowing river, the Qarqan River.

**Table 3.** M–K test results and center of gravity shifts for the watershed area of Lake Taitma, 2017–2022.

| M–K Test | Kendall's Tau | Alpha | *p*-Value | Hypothesis H$_0$ |
|---|---|---|---|---|
| — — | −0.733 | 0.1 | 0.060 | Reject |
| Year | Lon (°) | Lat (°) | Moving direction | Moving Distance (km) |
| 2017 | 88.164 | 39.511 | — — | — — |
| 2018 | 88.163 | 39.509 | southwest | 0.316 |
| 2019 | 88.154 | 39.509 | southwest | 0.961 |
| 2020 | 88.150 | 39.510 | northwest | 0.439 |
| 2021 | 88.116 | 39.513 | northwest | 3.811 |
| 2022 | 88.099 | 39.518 | northwest | 1.981 |

Note: The null hypothesis H$_0$ in the M–K test means that there is no trend in the sequence data.

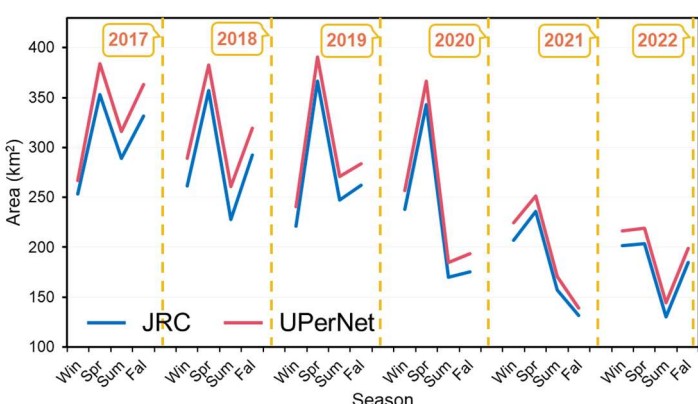 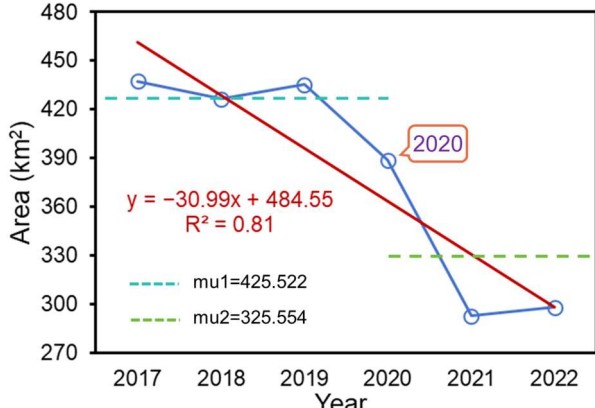

**Figure 7.** Seasonal and inter-annual variations in water area of Taitma Lake from 2017 to 2022.

### 3.3. Climatic Influences on Water Surface Dynamics in Taitma Lake

To further elucidate the climatic factors influencing the variability of water surface area in the Taitma Lake region, temperature, precipitation, and evapotranspiration data from the Ruoqiang and Qimo meteorological stations were compiled and analyzed. This analysis examined the meteorological elements affecting the surface water resources of Taitma Lake (Ruoqiang County) and the Qarqan River (Qiemo County) (Figure 8). The results show that Ruoqiang and Qiemo counties have scarce precipitation and high evapotranspiration, with precipitation primarily concentrated in the summer months. Over the past 6 years, the annual average precipitation amounts of Ruoqiang and Qiemo counties were 19.57 mm and 22.75 mm, respectively, the annual average temperatures were 11.81 °C and 11.68 °C, respectively, and the annual average evaporation amounts were as high as 1552.29 mm and 1826.83 mm. Between 2017 and 2022, these three types of meteorological factors essentially fail to demonstrate any clear inter-annual variation trend (the absolute value of the growth curvature is typically much smaller than 0.1), except for significant fluctuations in the growth curve of precipitation caused by precipitation anomalies in specific years. In particular, the evaporation data in Qiemo county have clearly indicated a downward trend over the past six years (growth curvature is −0.58), indicating that changes in climate factors over the past six years have had a significant positive impact on the growth of surface water resources in the Qarqan River Basin. Additionally, to express the relationship more clearly between climate elements and the water area of Taitma Lake, a Pearson correlation analysis was conducted. The results showed a negative correlation between the two, but neither passed the Pearson correlation test (Table 4). However, these factors alone do not account for the spatiotemporal changes in the water area of Taitma Lake over the past six years.

**Table 4.** Pearson correlation test results between the water area of Taitma Lake and various meteorological elements.

| Lake Taitma | Parameters | Average Annual Temperature (°C) | Annual Precipitation (mm) | Annual Evaporation (mm) |
|---|---|---|---|---|
| | Pearson correlation | −0.232 | −0.052 | −0.439 |
| Area (km²) | Significance | 0.658 | 0.922 | 0.384 |
| | Number of samples | 6 | 6 | 6 |

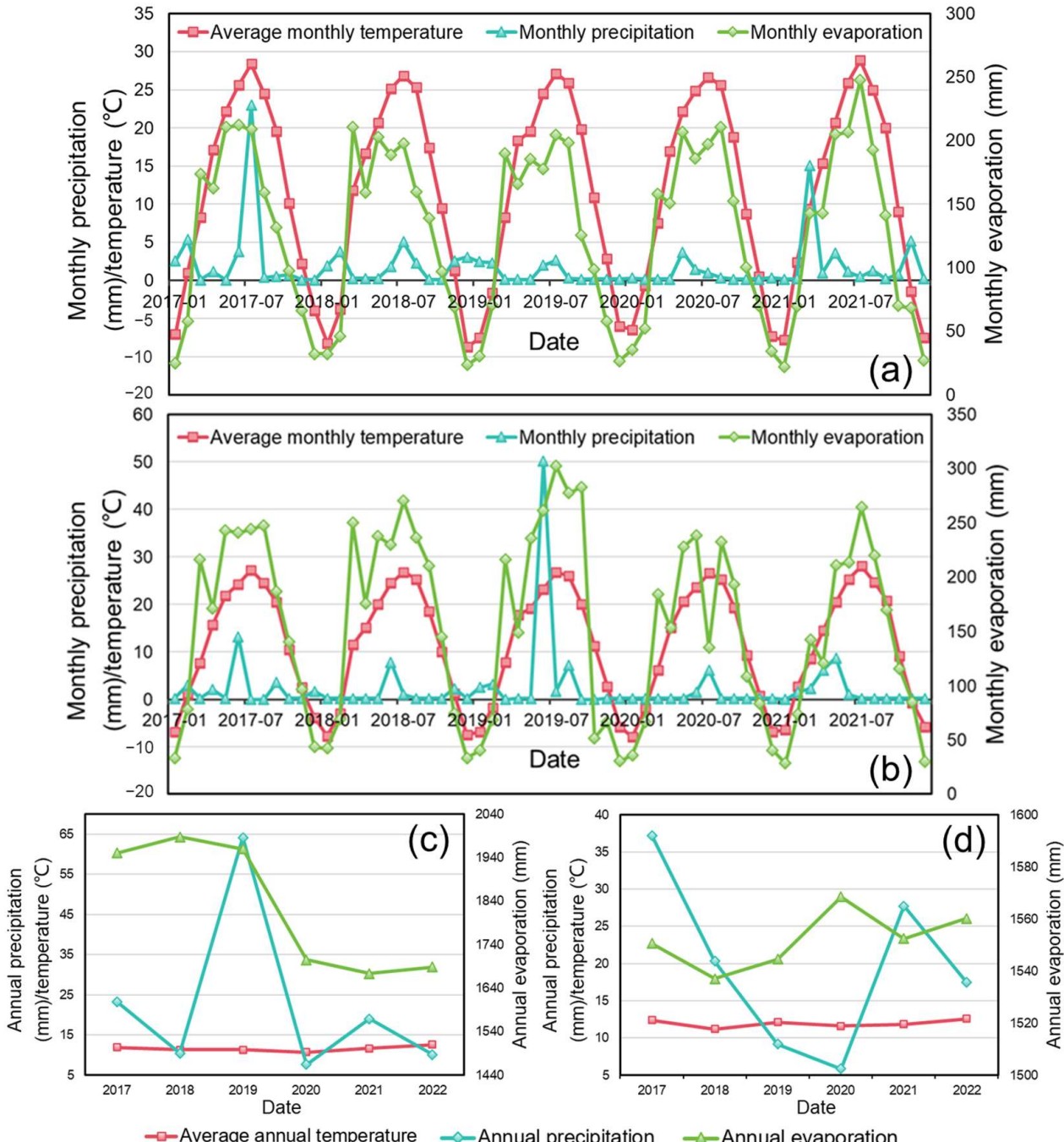

**Figure 8.** Monthly (**a**,**b**) and annual changes (**c**,**d**) of each meteorological factor in Ruoqiang and Zhimo counties, 2017–2022.

## 4. Discussion

### 4.1. Analysis of the Causes of Changes in Water Area of Taitma Lake

After the above analysis, we have drawn three brief conclusions: ① Taitma Lake's water area has gradually shrunk during the last six years; ② The Taitma Lake water area's center of gravity is shifting westward. This shift is likely due to a reduction in the lake's ability to receive water from the Qarqan River, resulting in the water supply being predominantly from the western side of the lake; ③ Between 2020 and 2022, profound changes are expected to occur in the two points mentioned above. However, climatic factors alone do not provide a sufficient explanation for these details observed. Consequently, we initially investigated the causes of seasonal variations in the water surface area of Taitma Lake.

Our analysis suggests that the peak in water surface area observed in March is likely attributable to the influx of water from snow and ice melt in the region. This phenomenon is also thought to be connected to the region's extremely high evapotranspiration throughout the summer. Additionally, when plants progressively progress towards the greening stage, the amount of water needed for development increases to its maximum, resulting in a reduction in summertime water area. The gradual recovery of the watershed area from summer to autumn is related to the ecological water transfer project of the Tarim River, which often takes place between August and November. From autumn to winter, as temperatures decrease, the lake's overall evaporation rate drops, plants go into their "hibernation period", and so does the water demand. Although the watershed area relatively increases compared to autumn, this is not an absolute increase. We have to take into account the amount of ecological water conveyance projects in the autumn of a specific year, as well as the amount of irrigation in agriculture, which may have led to the variations in the annual water area of the Taitma Lake from the autumn into the winter pattern [37].

Subsequently, our research on the Qarqan River revealed that the new Dashimen Water Conservancy Hub Project, constructed upstream, offers comprehensive benefits such as flood control, power generation, irrigation, and water supply. This project has effectively alleviated the seasonal water shortage in the irrigation area and has significantly contributed to local economic and social development [38,39]. The project began lowering and storing water on 20 September 2021 and was first put into spring production in 2022; prior to storage, this area was essentially dry. Based on PS satellite images, we monitored the water surface area of the Dashimen Water Conservancy Project during 2021–2022 (Figure 9). The results indicate a rapid increase in the water surface area since the beginning of water storage; the maximum water surface area in 2021 was recorded in October at 1.53 km$^2$, and in 2022, it peaked in August at 2.71 km$^2$. Since its completion, the water area of Taitma Lake has significantly contracted, with a notable inflection point in 2020, indicating a strong negative correlation between the water area of Taitma Lake and the water storage area of the Dashimen reservoir (Figures 7 and 9). Additionally, considering the substantial impact of the ecological water transfer project on the spatial distribution pattern of waters in the study area, we collected the total amount of water transfers from the lower Tarim River water transfer project to Taitma Lake over the years from 2017 to 2022 (Figure 10). The results show that the recharge volume of the ecological water transfer project in the lower Tarim River over the last six years exhibited a clear two-phase change: a sharp decrease in ecological recharge volume from 2017 to 2020, with an average annual decrease of $3.04 \times 10^8$ m$^3$. In contrast, climatic changes in Ruoqiang and Qiemo counties, as well as in the Taitma Lake and Qarqan River Basin, were not significant. Therefore, the gradual decrease in the basin area of Taitma Lake, averaging 12.2 km$^2$ per year during this period, can be attributed to a modest positive impact on the growth of surface water resources. During the period from 2020 to 2022, the amount of water replenishment slowly increased at an average annual rate of $0.57 \times 10^8$ m$^3$. On the assumption that there was no substantial change in the climate of the study area, the water area of Taitma Lake decreased dramatically (at an average annual reduction of 45.55 km$^2$). It can be assumed that the completion of the Dashimen Reservoir, when combined with the water storage time of the Dashimen Water Conservancy Project in the upper reaches of the Qarqan River, is the primary cause of the decrease in the water area of Taitma Lake between 2020 and 2022. This more adequately explains the 2020–2022 result, where the center of gravity of water distribution in the study area moved significantly westward during the year.

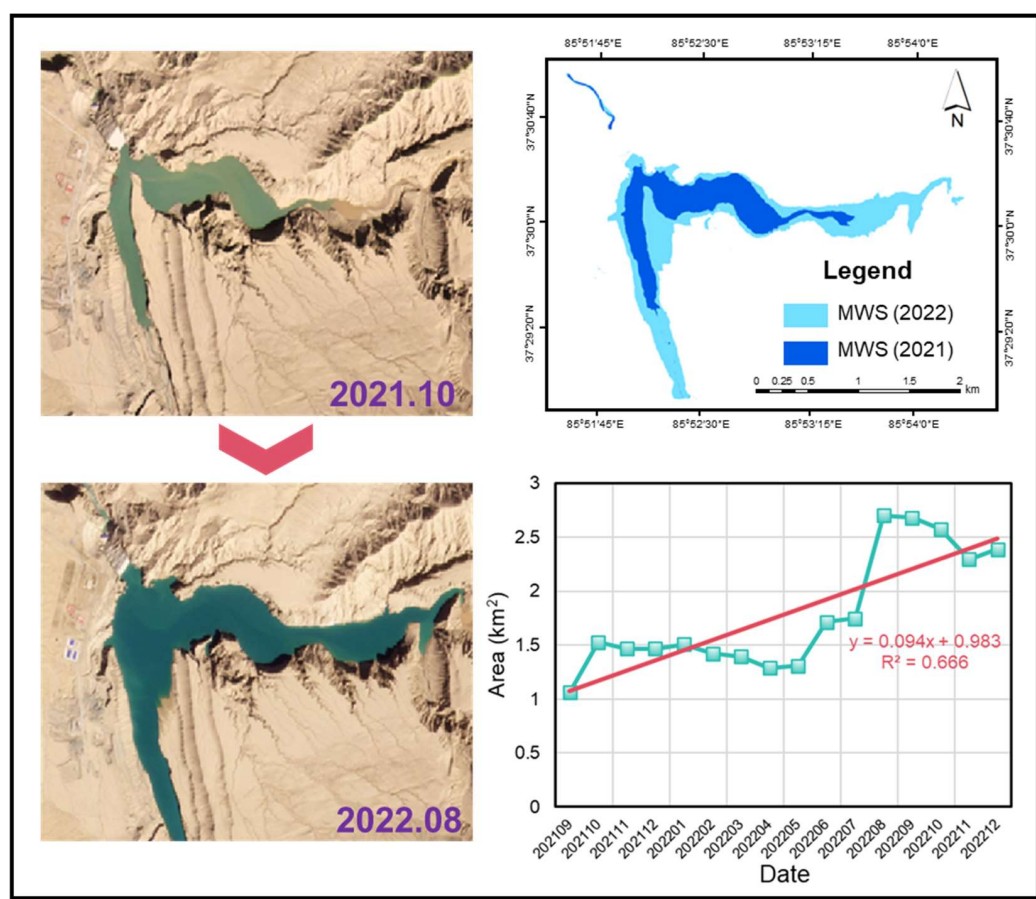

**Figure 9.** Changes in water surface area of Dashimen Water Conservancy Project from 2021 to 2022 (MWS means the maximum surface water area).

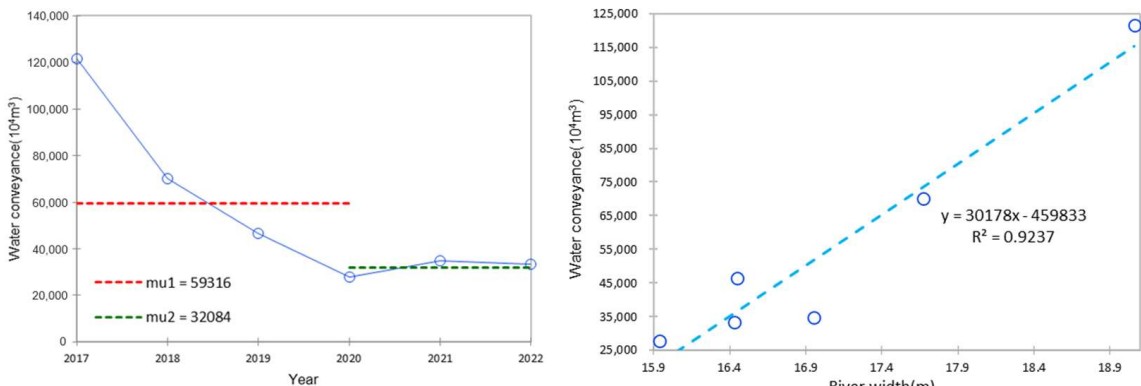

**Figure 10.** Changes of ecological water delivery in the lower reaches of the Tarim River from 2017 to 2022 and linear fitting results with S1 river width data.

In order to quantitatively describe the changes in the Tarim River's water supply to Taitma Lake over the past six years, we intercepted a section of the river (S1) at the entrance of the Tarim River on the east side of Taitma Lake and calculated the annual water surface distribution in the study area using GEE. The final river width of S1 is determined by extracting data pixel by pixel and calculating its average value. Simultaneously, by observing PS satellite images, we documented the frequency changes of water occurrence in S1 within the year (Figure 11). The findings reveal that over the past six years, the Tarim River, which has been impacted by water delivery regulation in its upper reaches, has demonstrated that the months of S1 with water have consistently occurred, fixed primarily

in March–April every year (when there is less water, mostly derived from precipitation), and in September–November (when there is a lot of water, mostly derived from ecological water transport). The frequency of water occurrence has not changed significantly, while the river width and ecological water transport exhibit highly similar change curves and correlation characteristics ($R^2$ = 0.924), indicating that the ecological transport volumes effectively represent how much water the Tarim River replenishes Taitma Lake. Therefore, the conclusion drawn from our analysis is relatively reliable: from 2017 to 2022, the decrease in the water area of Taitma Lake has been minimally influenced by climate change and is primarily affected by human activities. From 2017 to 2020, the water area of Taitma Lake was gradually reduced mainly due to the reduction in the inflow from the Tarim River, but the reduction rate was relatively small. From 2020 to 2022, the water area of Taitma Lake decreased substantially due to the Dashimen Water Conservancy Project, which resulted in significant water storage at the lower sluice.

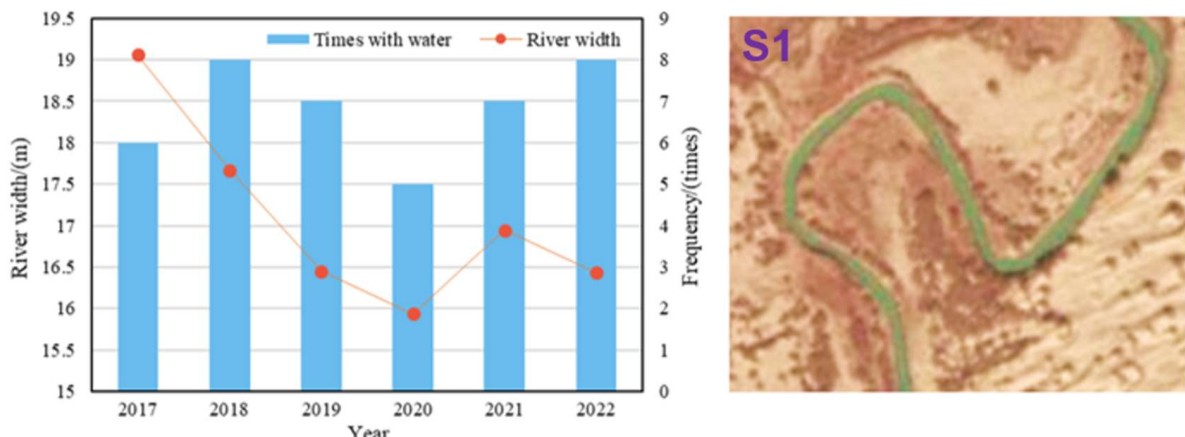

**Figure 11.** Changes in the width of the S1 section of the river and the frequency of water in the years from 2017 to 2022.

### 4.2. Implications of Water Area Variability in Taitma Lake

As the tail lake of an important river in the Tarim River Basin, Taitma Lake plays a crucial role in regional water ecological health [40]. It effectively blocks the convergence of the two major deserts and is located at the intersection of the 218 and 315 national highways. It also serves as a vital aquatic animal habitat and species gene pool in southern Xinjiang [19,41]. Regardless of whether it is a wet or dry year, maintaining a certain water area in the Taitma Lake region is of great strategic importance. The current management status is as follows: after years of ecological water transfers, the water area of Taitma Lake has gradually recovered, and the water environment has markedly improved [42]. However, owing to the climate in the area, the extension of the lake area also implies that the evaporation of the lake surface inadvertently increases the amount of ecological water transfer to the downstream, resulting in ecological water waste due to evaporation and tail lake leakage. Simultaneously, if the pursuit of maintaining the area of Taitma Lake is bound to seriously affect the upstream and middle reaches of agricultural production and domestic water, especially in drought years or regions, this could exacerbate the contradiction of water resources, resulting in a "high cost of replenishment, and low water use efficiency" situation. Therefore, the identification of the suitable surface of Taitma Lake can not only assist the local government to achieve more efficient ecological water transfer, but also provide an important theoretical basis for the sustainable restoration of the ecological environment in the Taitma Lake area.

Furthermore, based on the findings of this paper, the authors concluded that although the completion of the Dashimen Water Conservancy Project resulted in a significant short-term decrease in the watershed area of the downstream Taitma Lake, this may be advantageous in the long run as it can significantly lessen the pressure on water transport

and use in the middle and upper reaches of the Tarim River. As a result, regulating the amount of water provided to Taitma Lake by the Tarim and Qarqan Rivers is critical to maintaining an acceptable water surface in the future.

## 5. Conclusions

(1) In the water body extraction experiments, the deep learning model outperforms the conventional water body index method by a significant margin. UPerNet is the most successful network model in this study, with overall and fine water body-specific accuracies of 96.0 and 90.0%, respectively, and respective F1 values of 96.6 and 90.2%.

(2) The water area of Taitma Lake has clearly displayed seasonal variations between 2017 and 2022, reaching its peak value in March. Its water area has been significantly declining over the past six years at a rate of 31.12 km$^2$ per year on average, leading to the noticeable two-phase alterations (2017–2020 and 2020–2022) and a shift in the center of gravity of the water distribution to the northwest.

(3) The influence of human activity played a major role in the Taitma Lake watershed area's reduction during the past six years. Among them, the decrease in ecological water transfer from the upper Tarim River in 2017–2020 was the main reason for the decrease in Taitma Lake's watershed area, while the completion of the Dashimen Hydraulic Hub Project impoundment on the upper Qarqan River was the main reason for the exponential and sudden decrease in its area in 2020–2022.

In summary, our research leverages deep learning models to perform high-precision extractions of water bodies from freely accessible high-resolution PS imagery, providing a detailed spatiotemporal analysis of aquatic dynamics in the Taitma Lake region. This analysis enhances our understanding of the impacts of natural fluctuations and human interventions on semi-arid landscapes, offering valuable insights for environmental management and conservation strategies. Furthermore, this study introduces a method for extracting water bodies using high-resolution spatiotemporal imagery, demonstrating significant potential for application in similar ecosystems globally. Once data constraints are alleviated, this technique could be expanded to larger spatial scales and finer temporal resolutions, improving the analysis of micro-scale aquatic dynamics in arid and semi-arid regions worldwide. These capabilities are crucial for advancing the management of water resources in environmentally sensitive areas.

**Author Contributions:** Conceptualization, F.Z. and S.L.; formal analysis, S.L.; data processing, Y.W., C.F., X.L., N.W. and X.K.; writing—original draft preparation, Y.W.; writing—review and editing, S.L. and H.O.I.; visualization, Y.W.; supervision, F.Z. and S.L.; funding acquisition, S.L. All authors have read and agreed to the published version of the manuscript.

**Funding:** This research was funded by the National Key Research and Development Program of China (2021YFE0117800), the International Research Centre of Big Data for Sustainable Development Goals (CBAS) [grant number CBAS2023SDG002], and the National Natural Science Foundation of China (42171283).

**Data Availability Statement:** The original contributions presented in the study are included in the article, further inquiries can be directed to the corresponding author.

**Acknowledgments:** We thank the Tarim River Basin Management Bureau for the ecological water transfer data and the China Meteorological Administration for the meteorological station data in Ruoqiang and Qimo.

**Conflicts of Interest:** The authors declare no conflicts of interest.

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
