# Peer review of "Changes in the Water Area of an Inland River Terminal Lake (Taitma Lake) Driven by Climate Change and Human Activities, 2017–2022"

_remotesensing, doi:10.3390/rs16101703_

Round 1
Reviewer 1 Report
Comments and Suggestions for Authors
After reviewing the manuscript titled "Changes in the Water Area of an Inland River Terminal Lake (Taitma Lake) Driven by Climate Change and Human Activities, 2017-2022," I found that the paper compares several deep learning methods for assessing the accuracy of water area extraction in Taitma Lake and further evaluates the impact of climate change and human activities on the lake's area. The study is comprehensive and has the potential to be published in the field of Remote Sensing, but there are some issues that need to be addressed before publication.
The introduction lacks a section on current research progress. Comparing the accuracy of various deep learning models in extracting water area is a primary goal of the article, yet the introduction does not discuss the current progress in remote sensing methods for water area extraction or the contribution of this paper to the field.
In Lines 70-72, the authors mention that Sentinel satellite images with a 10m resolution are insufficient for monitoring water area changes, yet the study still employs a coarser resolution (30m) dataset (JRC Monthly Water History, v1.4) for global monthly water surface distribution.
The content in Lines 321-336 is not part of your results and should be included in the discussion section.
In Figure 7, the lines should use two sets of colors with stronger contrast.
The content in Lines 344-350 should belong to the methods section.
The content in Section 4.1 represents your results and should not be placed in the discussion.
I suggest renaming Section 4.2 since the authors do not provide an optimal water area but discuss the potential impacts of changes in the water area. Additionally, the last sentence of this section should be removed.
Comments on the Quality of English LanguageThe English in the manuscript requires significant improvement, particularly in terms of academic language usage. Please seek revisions and refinements from English experts within the same field. If possible, consider adding their names to the list of authors.
Author Response
Dear reviewer,
Thank you very much for your valuable comments and suggestions for us to improve the paper, based on your comments we have strictly checked the paper word by word according to the reviewer’s comments and try our best to make the responses. For details, please check the response given in the uploaded file. Thanks.
Best regards,
Wang Yong

Reviewer 2 Report
Comments and Suggestions for Authors
I have carefully reviewed your manuscript titled "Changes in the water area of an inland river terminal lake (Taitma Lake) driven by climate change and human activities, 2017–2022." Overall, I find the study to be insightful and potentially valuable for understanding the dynamics of water area changes in Taitma Lake. The utilization of Planetscope (PS) satellite images and deep learning models for water body extraction presents an innovative approach, and the observed accuracy of up to 96.0% is noteworthy. However, a few aspects of the manuscript require further attention to ensure the robustness and clarity of the findings. In particular, I recommend addressing mild concerns related to introduction elaboration and clarification, attribution of changes in water area, and discussion of uncertainties. Additionally, enhancing the scope of the methodology and implications of the findings on a global scale would enrich the manuscript's contribution to the fields of water resource management and ecological protection.
Author Response
Dear reviewer,
Thank you for your thorough review and the insightful comments on our manuscript titled "Changes in the water area of an inland river terminal lake (Taitma Lake) driven by climate change and human activities, 2017–2022." We appreciate your recognition of the innovative approach we employed using Planetscope (PS) satellite images and deep learning models for high-precision water body extraction, and are pleased that the accuracy achieved is noted as noteworthy.
We acknowledge the areas identified for improvement and agree that addressing these will enhance the manuscript's robustness and clarity. We are currently working to address the concerns raised and will revise our manuscript accordingly to ensure that our findings are presented as clearly and robustly as possible. For details, please check the response given in the uploaded file
Thank you once again for your valuable feedback, which has undoubtedly helped strengthen our work.
Best regards,
Wang Yong

Reviewer 3 Report
Comments and Suggestions for Authors
The manuscript presents a comprehensive study on the changes in the Taitma Lake area from 2017 to 2022, focusing on the impact of climate change and human activities. The authors utilize high-resolution PS satellite imagery and deep learning models to monitor changes in the lake area, considering both natural ecological responses and human-induced alterations.
The research methodology is well-designed and effectively implemented. By utilizing advanced techniques such as UPerNet for water extraction from satellite images, the authors achieve promising results, particularly in accurately identifying water bodies. The comparison between deep learning models and traditional methods highlights the superiority of the proposed approach, providing valuable insights into the effectiveness of different techniques for water extraction.
Furthermore, the study incorporates various data sources including meteorological data, remote sensing monitoring data, and auxiliary data, ensuring a comprehensive analysis of the factors influencing changes in the lake area. The use of Mann-Kendall trend tests and Pearson correlation analysis adds robustness to the findings, enabling a thorough investigation of spatiotemporal trends and correlations.
Overall, the manuscript contributes significantly to the understanding of lake dynamics in arid and semi-arid regions, demonstrating the potential of high-resolution satellite imagery coupled with deep learning models for monitoring environmental changes. The findings have implications for water resource management and ecosystem conservation in similar regions worldwide.
Author Response
Dear reviewer,
Thank you very much for your thorough review and the constructive comments provided on our manuscript. We are grateful for your recognition of the comprehensive nature of our study on the changes in the Taitma Lake area from 2017 to 2022 and the impact of climate change and human activities.
We appreciate your positive feedback on the methodology and the use of advanced techniques such as UPerNet for water extraction from high-resolution PS satellite imagery. It is encouraging to hear that our efforts to compare deep learning models with traditional methods have provided valuable insights, as noted in your comments
Sincerely.
We aim to continuously improve our work based on your insightful feedback and are committed to contributing further to the field. Thank you once again for your supportive and detailed review.
Best regards,
Wang Yong
Reviewer 4 Report
Comments and Suggestions for Authors
The study sounds great, however, some improvements are required to enhance your paper. Please see comments below:
1- Introduction: Discuss advantages and disadvantages of your approach. This is very important in distinguishing your approach from others.
2- Line 102: Fix the caption for figure 1
3- Study Area: Refer to figure 1 in-text in the section “Study Area”. Make sure that all figures in your study are properly (in-text) referenced.
4- Data Sources: justify why you selected the study period 2017-2022
5- Line 163: Please justify selecting 512x512 pixels window
6- Accuracy evaluation:
7- Lines 316-320: Try to re-write for simplification and add a justification as well of why there is a limitation on PS imageries. was it because cloud covers or no more images were available?
8- Figure 8: Some improvements could be made to enhance clarity
9- Same comment above for figure 9 & 10.
10- Conclusion: Add more text about the perfection pf your approach. In other words, advantages of selecting your methodology to conduct this study.
Comments on the Quality of English LanguageNo major English issues were detected.
Author Response

(The authors gave the same response as above.)

Round 2
Reviewer 1 Report
Comments and Suggestions for Authors
The authors have adequately addressed the commte I previously raised. I now consent to accepting this manuscript.
Comments on the Quality of English LanguageThe manuscript's English writing is generally acceptable, but attention should be paid to the accurate use of academic terminology. It is advisable for the authors to seek in-depth proofreading from foreign experts in the relevant field to ensure precision and clarity.
Author Response
Dear reviewer,
First of all, we sincerely thank you for reviewing our manuscript and your valuable manuscript comments. Your feedback is essential for us to improve the manuscript.
Based on your suggestions, we have made comprehensive and aesthetically pleasing revisions to our work. We have paid special attention to improving the linguistic quality of Strawberry's pronunciation and accuracy through coordinated touch-ups, improved grammatical structure and fluency of expression. We are certain that these efforts will significantly enhance the overall quality of the manuscript and communicate our findings more effectively.
To facilitate your review, we have uploaded two versions of the manuscript: a revised manuscript with modification marks showing all the changes made based on your suggestions, and a clean version for final review. Please refer to the revised manuscript with modification marks for a detailed revision comparison.
We sincerely hope that the manuscript will now meet your standards for publication. Thank you again for your thorough review and helpful suggestions. Your time and consideration are greatly appreciated.
Best regards,
Wang Yong
